

**Validation of INSAT-3D sounder data with in-situ measurements and other**
**similar satellite observations over Indian region**
**M. Venkat Ratnam*, A. Hemanth Kumar, and A. Jayaraman**
National Atmospheric Research Laboratory, Gadanki, India
*vratnam@narl.gov.in ; Phone: +91-8585-272123; Fax: +91-8585-272018





**Abstract**
To date, several satellites measurements are available which can provide profiles of
temperature and water vapor with reasonable accuracies. However, temporal resolution remained
poor, particularly over topics, as most of them are polar orbiting. At this juncture, launch of
INSAT-3D (Indian National Satellite) by the Indian Space Research Organization (ISRO) on 26
July 2013 carrying multi-spectral imager covering visible to long wave infrared region made it
possible to obtain profiles of temperature and water vapor over Indian region with higher
temporal and vertical resolutions and altitude coverage besides the other parameters. The initial
validation of INSAT-3D data is made with the high temporal (3 h) resolution radiosonde
observations launched over Gadanki (13.5°N, 79.2°E) during a special campaign and routine
evening soundings obtained at 12 UTC. We also compared INSAT-3D data with the radiosonde
observations obtained from 34 India Meteorological Department stations. Comparisons were also
made over Indian region with data from other satellites like AIRS, MLS and SAPHIR and ERA-
Interim and NCEP re-analysis datasets. INSAT-3D is able to show a better coverage over Indian
region with high spatial and temporal resolutions as expected. Good correlation in temperature
between INSAT-3D and in-situ measurements is noticed except in the upper troposphere and
lower stratospheric region (positive bias of 2-3K). There exists mean dry bias of 10-25% in
relative humidity. Similar biases are also noticed when compared to other satellites and re-
analysis data sets. INSAT-3D shows large positive bias in temperature above 25°N in the lower
troposphere. Thus, caution is advised in using this data at those places for tropospheric studies.
Finally it is concluded that temperature data from INSAT-3D is of high quality that can be
directly assimilated for better forecast over Indian region.
***Key words:*** Temperature, relative humidity, INSAT-3D, radiosonde, MLS, AIRS, reanalysis.



## 1. Introduction

Temperature and water vapor play an important role in deciding the thermodynamic state of the atmosphere as they are considered as feedback parameters which alter the radiation and moist dynamics of the atmosphere. The stability of the Earth's atmosphere (troposphere and stratosphere) depends on the density of the air parcel at any particular altitude. The density of the air parcel depends on the amount of water vapor present in it and also its temperature. The water vapor is a highly varying parameter which is mainly responsible for precipitation and all other weather systems. It is the source of the latent heat which is released into the atmosphere during the cloud formation. It also dominates the structure of diabatic heating of the Earth's atmosphere (Trenberth et al., 2005; Trenberth and Stepaniak, 2003a; 2003b).These parameters vary in time and as well as in space (both vertically and horizontally) throughout the atmosphere.

Profiles of temperature (T) and relative humidity (RH) water vapor are traditionally obtained from the in-situ conventional radiosonde measurements which have high vertical resolutions and accuracies. However, they have limited spatial and temporal coverage. For this reason, the satellites are considered as the best source of information for obtaining these parameters which provide observations on a global scale and with improved temporal resolution based on the orbit in which the satellite is present. Among several satellites, Atmospheric Infrared Sounder (AIRS), Microwave Limb Sounder (MLS) and GPS Radio Occultation provide profiles of temperature and water vapor with reasonable accuracies. Recently Sounder for Atmospheric Profiling of Humidity in the Inter-tropical Regions (SAPHIR) onboard Megha Tropiques has been introduced which provides profiles of RH in the tropical latitudes (Venkat Ratnam et al., 2013). They have good spatial coverage but the temporal resolution of these satellites is poor. At this juncture launch of Indian National Satellite System (INSAT)-3D in July



2013 has gained lot of significance due to its geostationary transfer orbit which provides profiles
of T and RH with high temporal resolutions, though restricted to Indian region only when
compared to other satellites mentioned above. This data is expected to play important role in
numerical weather prediction over Indian region. Before using this data for weather forecasting,
it is essential to validate with in-situ, similar satellite and re-analysis data sets.
In this report, we discussed the features of T and RH obtained from INSAT-3D sounder.
It adds a new dimension by providing continuous observations of T and RH over the Indian
region and thereby useful in monitoring the Earth's weather systems continuously. In the first
section we compared the broad features of T and RH obtained from INSAT-3D with the other
satellite observations. It is followed by the validation of INSAT-3D data with high resolution
radiosonde launched during a special campaign (Tropical Tropopause Dynamics Campaigns)
(Venkat Ratnam et al., 2014) and routine evening soundings over Gadanki (13.5°N, 79.2°E), a
tropical station in the southern peninsular India. We also compared this data with the India
Meteorological Department (IMD) network of radiosonde consisting of 34 stations over Indian
region. In this context it is worth to quote Mitra et al. (2015) where they compared INSAT-3D
data obtained from January 2014 to May 2014 with 10 GPS stations of IMD. However, their
work is restricted up to 100 hPa only and for initial 5 months. In the present work we extended
comparisons for complete 2 years (2014 and 2015) and up to 10 hPa. Further, the comparisons
are also made with other satellite observations like AIRS (Atmospheric Infrared Sounder),
Microwave Limb Sounder (MLS), and SAPHIR (Sounder for Atmospheric Profiling of Humidity
in the Inter-tropical Regions) and re-analysis data sets like ERA-Interim (European Center for
Medium Range Weather Forecasts ECMWF), NCEP (National Center for Environmental
Prediction).





## 2. Database

### 2.1. INSAT-3D

The INSAT-3D which is considered to be the advanced version of all the other INSAT series satellites is the meteorological satellite of ISRO launched from Kourou, French Guiana as a passenger payload along with AlphaSat / InmarSatI-XL, ESA/ InmarSat by the European launch vehicle named Ariane-5 VA-214 on 26 July 2013. It was positioned at 82°E over the equator at an altitude of 35,786 km from the surface of Earth in the Geostationary Transfer Orbit (GTO) with the main objectives of monitoring the earth and ocean continuously thereby providing the data dissemination capabilities. It also provides an operational, environmental and storm warning system to protect the life and property. It carries four payloads namely the multi-spectral imager (optical radiometer) which provides the high resolution images of the mesoscale phenomena and local storms mainly in the visible band, apart from imaging the whole earth disk in the shortwave Infrared, middle Infrared, water vapor and low thermal Infrared channels. The atmospheric sounder which has 19 channels in shortwave infrared, middle infrared, long wave infrared (18) and visible (1) channel measures the irradiance and provides the profiles of T, RH and integrated ozone over the selected land mass of the Indian region every hour and whole Indian ocean every six hours as show in Table 1.

This atmospheric sounder gives profiles of T and RH at 40 pressure levels (1000, 950, 920, 850, 750, 700, 670, 620, 570, 500, 475, 430, 400, 350, 300, 250, 200, 150, 135, 115, 100, 85,70,60, 50, 30, 25, 20, 15, 10, 7, 5, 4, 3, 2, 1.5, 1, 0.5,0.2, 0.1 hPa) for every one hour  at 10 km x 10 km in latitude and longitude resolutions covering 5-40°N and 60-100°E over India  region. The INSAT-3D sounder provides T and RH profiles along with the total columnar ozone from the Infrared radiances obtained in different absorption bands during the clear sky conditions. The



retrieval algorithm adopted for INSAT-3D sounder is same as that adopted for HIRS (High
resolution Infrared radiation sounder) and GOES sounder which are mainly based on the
retrieval algorithm of Hayden (1988), Ma et al. (1999) and Li et al. (2000).
**2.2. Radiosonde observations**
The processed and quality checked radiosonde data obtained from the Integrated Global
Radiosonde Archive (Durre et al., 2006) over Indian region at different locations (0-40˚ N, 60-
100˚E) during the period 2014-2015 are obtained. The observed unexpected sharp spikes in the
data are removed and the data values which are within the range $\pm 2\sigma$ are only considered for
comparison. Such stringent quality checked data obtained are utilized for comparing with the T
and RH obtained from INSAT-3D. The 34 locations of the radiosonde stations over the Indian
region (i.e., IMD stations) are shown in the Figure 1. The data from these IMD stations obtained
at 00 UTC are only used for comparison as the 12 UTC data during this period is very sparse.
Further, high altitude resolution GPS radiosondes (Meisei RS-11 G, Japan) that were
launched over Gadanki around 12 UTC are also used in the present study. Besides these routine
evening radiosonde launches, the radiosondes that were launched as a part of special campaign
between January 2014 and March 2014 over the same location are also utilized for comparison at
sub-daily scales. The sensors used for measuring the T and RH are thermistors and carbon
hygristors, respectively. The range of the T and RH measured by the sensors are -90 to +50 ° C
and 1-100 % with an accuracy of 0.5 K and 5-7 %, respectively (Basha and Ratnam, 2009;
Venkat Ratnam et al., 2014). During this campaign the radiosondes were launched for every 3
hours (11:30, 14:30, 17:30, 20:30, 23:30, 02:30, 05:30 and 08:30 IST) continuously for three
consecutive days. The entire radiosonde datasets are interpolated to the pressure levels of
INSAT-3D data.





**2.3. Other satellite observations**
**2.3.1. AIRS observations**
AIRS is one of the payloads on the NASA Earth Observing System satellite called
AQUA which is in a polar sun synchronous orbit revolving at an altitude of 705 km from the
Earth's surface with an orbital period 98.99 minutes. It completes approximately 14.5 orbits per
day and the separation between any two consecutive orbits near the equator is 2760 km. The
partner payloads along with AIRS onboard AQUA satellite are microwave instruments AMSU
and Humidity Sounder for Brazil. The satellite crosses the equator twice a day one being during
the ascending node at ~13:30 UTC and the other one being during the descending node at ~01:30
UTC. It is a high spectral sounder with 2378 channels measuring the IR radiances at wavelengths
in the range of 3.7–15.4 μm with a swath of 1,650 km and horizontal spatial resolution of 13.5
km at nadir (Aumann et al., 2003). We used the level 3 version 5 daily gridded data products
(Susskind et al., 2006) that are obtained from the IR radiances of AIRS sounder during 2014 and
2015. The level 3 data products (AIRS V5 L3) are obtained from the level 2 swath data where
the data of all the 15 orbits of the day are averaged together and the data has a latitudinal and
longitudinal resolution of 1° X 1° at 24 pressure levels for T starting from 1000hPa to 1hPa and 12
levels for RH from 1000hPa to 100hPa. Note that RH data is reliable in the first 8 levels from the
surface and up to 300 hPa (Waters et al., 2006).
**2.3.2. MLS observations**
MLS is one of the four payloads onboard NASA's EOS Aura satellite which is one
among the six satellites (OCO-2, GCOM-W1, AQUA, CLOUDSAT, CALIPSO, AURA) that
form the A-Train constellation. Similar to AIRS, MLS is also a polar orbiting sun synchronous
satellite (AURA) which is at ~705 km, scanning its view from ground to ~90 km at 55 pressure





levels with a global view covering from 82° S to 82° N by having ~15 orbits per day. It scans the
Earth's atmosphere for every 25 seconds and provides 240 scans per orbit. The details regarding
the MLS measurement technique, instrumentation are discussed by Waters et al. (2006). The
MLS measures the thermal emission of the earth through its limb viewing geometry at
microwave band centered near 118 GHz, 190 GHz, 240 GHz, 640 GHz and 2500 GHz whose
retrieval algorithm can be found from Livesey et al. (2006). We made use of the Level 2 version
3 temperature and water vapor data during the period 2014 and 2015 that was downloaded from
http://mirador.gsfc.nasa.gov. Note that water vapor from this instruments is more valid above
300 hPa only (Basha et al., 2013).
**2.3.3. SAPHIR observations**
SAPHIR is one of the four instruments onboard Megha Tropiques (MT) satellite which is
moving in a circular low inclination orbit at 20° with 14 orbits per day. It provides a cross track
scan of ±43° with a swath of 1705 km and resolution of 10 km at nadir. It is a passive remote
sensing microwave sounder which operates at 6 channels close to 183.31 GHz (±11.0, ±6.60,
±4.30, ±2.8, ±1.2) retrieving integrated RH of the entire troposphere from brightness temperature
at 1000-850 hPa, 850-700 hPa, 700-550 hPa, 550-400 hPa, 400-250 hPa and 250-100 hPa within
±30° latitudinal belt. The algorithms related to retrieval for the sounders of MT satellite are
discussed by Gohil et al. (2012). This data has been validated against similar satellites and
reanalysis data sets and found good except in level 1 (1000-850 hPa) (Venkat Ratnam et al.,
2013). We made use of the SAPHIR data for comparison which was downloaded from
www.mosdac.gov.in for the period 2014 and 2015.






### 2.4. Re-analysis datasets

### 2.4.1. ERA-Interim data

ERA-Interim is the advanced global atmospheric reanalysis which is produced by ECMWF. It provides gridded data products which include large surface parameters for every 3 hours and upper air parameters covering troposphere and stratosphere for every 6 hours starting from 1979 onwards. The data products are obtained from the model through sequential data assimilation method where the models are fed with the available observations to forecast the evolving state of the global atmosphere. The configuration and performance of the ERA-Interim reanalysis is explained clearly by Dee at al. (2011). It is even considered as the latest and most advanced global assimilation scheme which can predict the atmosphere at the nearest accuracy to what is theoretically possible (Simmons and Hollingsworth, 2002; Simmons et al., 2007). These data products are available over the entire globe at different latitude and longitude resolutions and for 37 pressure levels from 1000 hPa to 1 hPa. We have made use of $1° \times 1°$ data products of T and RH for the period 2014 and 2015.

### 2.4.2. NCEP/NCAR data

This data set is a joint product of National Centers for Environmental Prediction (NCEP) and National Center for Atmospheric Research (NCAR). Similar to ERA-Interim data this is also provides gridded data which is available from 1948 onwards. NCEP data represents the state of Earth's atmosphere by incorporating the global historical observations and the output of global numerical weather prediction (NWP) model (Kalnay et al., 1996). These data products are available all over the globe at $2.5° \times 2.5°$ latitude –longitude resolution at 17 pressure levels starting from 1000 hPa to 10 hPa for temperature and 8 pressure levels for RH from 1000 hPa to 300 hPa. We made use of this data for T and RH during the period of 2014 and 2015.



## 3. Results and discussion

### 3.1. Spatial variation of T and RH over Indian region



In this section, the observations of the advanced ISRO geostationary INSAT-3D satellite
sounder which provides continuous observations over land and ocean of Indian region are
discussed as they are very important in weather forecasting. The continuous observations of the
sounder are very much important as these observations can be introduced and combined with
model output for a better forecast of the Earth's atmosphere. Before it is used for any scientific
purpose, it is essential to compare / validate with other similar data sets. Figure 1 shows the
spatial variation of T and RH over Indian region at 850 hPa pressure level obtained from INSAT-
3D satellite on 2 May 2015 (averaged over a day). White patches show the non-availability of
the data due to topography (Himalayan Mountains). Higher temperatures of about 5-6 K over the
main land mass than surrounding sea can be noticed. On the contrary, very low values of RH
over the land mass than surrounding ocean can be noticed.
The simultaneous observations from MLS and AIRS over the Indian region obtained
around 13:30 IST (i.e., ascending node for AIRS and MLS) on the same day is considered for
comparison. Spatial variation of T and RH over Indian region observed at 500 hPa pressure level
from INSAT-3D, MLS and AIRS satellites on 2 May 2015 around 13:30 IST is shown in Figure
2. Spatial variation of T and RH over Indian region obtained from ERA-Interim and NCEP at the
same pressure level at 06 UTC (11:30 IST) is also shown. In general, though major features in
the spatial variation of T resembles among different satellites and re-analysis data sets, however,
large difference in the RH is noticed. Particularly AIRS show large RH variations over Bay of
Bengal (BoB) and Himalayan region when compared to other two satellites. Similar high
variation in RH is also seen by ERA-Interim (Figure 2i). Very low RH values in the central India



and toward west in all the satellite observations can be noticed. The quantitative difference
between INSAT-3D and other satellite measurements and re-analysis data sets will be discussed
in later sections.

**3.2. Comparison between INSAT-3D and radiosonde observations at sub-daily scales**

INSAT-3D sounder provides the profiles of T and RH over Indian region for almost

every hour. It is desirable to compare these profiles at different timings of the day which is
difficult to do with existing polar satellites. Thus, we compared the INSAT-3D profiles of T and
RH with radiosonde observations obtained over Gadanki and also using IMD network of
radiosondes. It is well known that the most common and widespread in-situ instruments for
providing accurate profiles of T and RH are radiosondes. However, accurate measurements of
RH are found to be a challenging task with the help of radiosondes in the upper troposphere and
lower stratosphere where the concentration of water vapor is very low. In addition to this there
exists radiation error in temperature measurements as explained by Luers and Eskridge, (1998)
and Wang et al. (2003). We have used high accuracy and vertical resolution radiosonde over
Gadanki that were launched for every 3 hours continuously for three consecutive days, during a
special campaign called Tropical Tropopause Dynamics campaign (TTD) (Venkat Ratnam et al.,
2014) conducted over Gadanki between January 2014 and March 2014. This data is used to
validate the INSAT-3D measurements at sub-daily scales.

The radiosonde data obtained during the TTD campaigns are interpolated to the pressure

levels of INSAT-3D for the similar hours whenever observations are available. Typical temporal
variation of T and RH obtained from radiosonde launched over Gadanki during one of the TTD
campaigns conducted from 27 January 2014 to 30 January 2014 is shown in Figure 3. Data
obtained from INSAT-3D for similar timings are also shown in the bottom panels. White patches





show the non-availability of the data. In general, similar diurnal variation in the T and RH
between radiosonde and INSAT-3D can be noticed though the magnitude differs. Very cold
temperatures (~190 K) present near the tropopause region (100 hPa) are captured well by
INSAT-3D. The existence of high humidity layer at 300 hPa, persisting for more than a day, is
also captured well by the INSAT-3D. The T and RH over Gadanki obtained from INSAT-3D and
radiosonde are averaged over 3 days and the mean and standard deviation are shown in Figure
3(e) and 3(f), respectively.  From these profiles, no significant difference in the T can be noticed
but there exists underestimation in RH by INSAT-3D (assuming radiosonde as standard
technique). INSAT-3D shows a dry bias of 20-35% in RH when compared to radiosonde
observations. No significant day-night differences are noticed between INSAT-3D and
radiosonde observations.
**3.2. Comparison between INSAT-3D and radiosonde (IMD and Gadanki) observations**

We also compared INSAT-3D measurements obtained during 2014 and 2015 with the

radiosonde observations over the 34 IMD stations which are spread throughout the Indian region
whose locations are shown in the form of filled circles in Figure 1. Besides these, the routine
evening radiosonde observations launched around 12 UTC over Gadanki during 2014 and 2015
were also utilized for day-to-day comparisons. The radiosonde data of all the IMD stations are
interpolated to the pressure levels of INSAT-3D for uniformity. The correlation co-efficient
values obtained for T and RH between INSAT-3D and Gadanki radiosonde launched around 12
UTC and IMD radiosonde launched around 00 UTC over Indian region are obtained separately
for each day during the period 2014 and 2015. The correlation values are obtained for all the
levels in T whereas only up to 300 hPa in RH and is shown in Figure 4. A very high correlation
(>0.8) in T between INSAT-3D and IMD /Gadanki radiosonde is observed in the lower





troposphere (Figure 4a). However, correlation decreases above 700 hPa (850 hPa) between
INSAT-3D and Gadanki (IMD) radiosonde. There exists consistent correlation of more than 0.6
throughout all levels with Gadanki radiosonde but drastically decreases above 250 hpa in case of
IMD radiosondes. It is interesting to notice higher (lower) correlation below (above) 850 hPa
between Gadanki radiosonde and INSAT-3D. However, it is quite opposite in case of IMD
radiosonde for which reasons are not known. The correlation values of RH obtained between
INSAT-3D and Gadanki radiosonde is always higher (greater than 0.65) throughout the profile
than the correlation obtained between INSAT-3D and IMD radiosonde observations (less than
0.5) shown Figure 4b. Mitra et al. (2015) has reported similar correlations using 10 IMD stations
using 5 months (January 2014- May 2014) of the data. However, their work is restricted up to
100 hPa due to frequent balloon burst of IMD radiosondes at that altitude. In the present study
we report up to 10 hPa and also using complete two years of the data for Gadanki location. The
observed good correlation (0.6-0.7) between INSAT-3D RH and Gadanki radiosonde RH may be
attributed to the improved RH sensor used in Meisei radiosondes that were used over Gadanki.
Further, to quantity the differences between INSAT-3D and Gadanki radiosonde, we
discuss the fractional difference at all levels between routine radiosondes launched around 12
UTC over Gadanki and INSAT-3D T over the same site during the period 2014 and 2015. The
fractional difference of T for each day is calculated separately and then averaged over 2014 and
2015. The balloon bursting altitude of the radiosonde is also estimated for those which are
utilized in estimating the fractional difference. The fractional difference of T and RH and balloon
bursting altitude are shown in Figure 5. It is clear from the figure that the difference is very less
in the troposphere (~0.5 K). The mean fractional difference in the troposphere is less than 0.5 K,
and it is about 1 K in the upper troposphere and lower stratosphere. However, positive bias





(INSAT-3D showing higher temperatures) of 2-3 K is noticed (shown as standard deviations) in
day-to-day differences in INSAT-3D. When we segregated season wise fractional differences,
higher fractional difference during monsoon season is noticed (figure not shown) mainly due to
less number of matches between INSAT-3D and radiosonde due to over sky. Most striking
feature to be noticed is the consistent positive bias of 1% (~2 K) in T in the upper troposphere
and lower stratosphere. The mean fractional difference in RH shown in Figure 5b reveals 20-
30% dry bias in INSAT-3D when compared to radiosonde. Standard deviations show dry bias of
40-60% in day-to-day comparison of RH between INSAT-3D and radiosonde. Thus, from figure
5, it is clear that INSAT-3D is able to provided T measurements with high accuracies but huge
dry bias is observed in RH. Thus, caution is advised while using RH data from INSAT-3D.
**3.2. Comparison between INSAT-3D and other satellite and re-analysis data**
The T and RH measured from the radiances of 19 channels of INSAT-3D sounder are
compared with that are obtained from other satellites like AIRS, MLS and SAPHIR (only RH)
during the period 2014 and 2015. Besides the satellite observations, re-analysis datasets like
ERA-Interim and NCEP are also utilized for comparing the data obtained from INSAT-3D. The
T measurements obtained from AIRS, MLS, ERA-Interim are converted to a spatial resolution of
1° X 1° in latitude and longitude. The 1° X 1° gridded AIRS and MLS T measurements are
interpolated to 40 pressure levels of INSAT-3D. Whereas, the INSAT-3D data is converted to a
spatial resolution of 2.5°X 2.5° to compare with the T obtained from NCEP. The difference in T
between INSAT-3D and AIRS and MLS are estimated for each day, whereas it is estimated for
every six hours between INSAT-3D and ERA-Interim and NCEP. The zonal mean latitudinal
difference of T between different satellites and re-analysis datasets is obtained for each day and
then averaged for 2014 and 2015 which is shown in Figure 6. Note that the differences that are



greater than 1K are only shown in this figure. In general, the difference in T between INSAT-3D
and other satellite and reanalysis data sets lies within ± 1 K and extends to 2 K in the UTLS
region. Above 25°N, INSAT-3D shows positive bias of more than 4 K up to 300 hPa compared
to AIRS but up to 700 hPa with rest of the data sets. Consistent positive bias of 2-3 K in the
UTLS region can be noticed in INSAT-3D particularly compared with other satellite
measurements. Above 4 hPa, consistent negative bias of more than 3 K is noticed in INSAT-3D
when compared to other data sets. In general, less difference between INSAT-3D and NCEP is
noticed than ERA-Interim. Thus, the difference in T between INSAT-3D and other datasets is
least in the lower and mid troposphere below 25°N, whereas it increases in the lower troposphere
above 25°N.

The RH data obtained from AIRS, MLS, ERA-Interim are converted to a spatial

resolution of 1°X 1°in latitude and longitude and then interpolated to the first 21 pressure levels
of INSAT-3D. To compare the INSAT-3D RH data with NCEP RH data, the RH data obtained
from INSAT-3D is converted to the actual resolution of NCEP, i.e., 2.5°X2.5° latitude and
longitude grids. Note that information on RH data obtained from NCEP is present only up to 300
hPa, MLS from 300 hPa and above, whereas RH from AIRS and ERA-Interim is considered up
to 100 hPa beyond which the concentration of water vapor is very low. But, the RH obtained
from SAPHIR in the troposphere is measured as integrated relative humidity at certain levels as
mentioned in the section 2. In order to compare the INSAT-3D RH data with SAPHIR RH, the
former is converted to the pressure levels of SAPHIR. The zonal mean latitudinal difference
between INSAT-3D and all other datasets is obtained as mentioned in the previous section and is
presented in Figure 7. In general, INSAT-3D shows a dry bias of 5-10% in the lower and mid
troposphere below 25°N when compared with AIRS (Figure 7a), ERA-Interim (Figure 7c) and





NCEP (Figure 7d) re-analysis datasets. However, it shows a dry bias of more than 10% when
compared with MLS RH (Figure 7b). Note that INSAT-3D also shows a wet bias around 700
hPa with all the datasets. A high dry bias in the lower troposphere above 25° N is observed
between INSAT-3D and AIRS, ERA-Interim and NCEP, whereas the bias in the same region is
less with MLS. The wet bias (~20%) between INSAT-3D and AIRS above 300 hPa is mainly
due to low accuracies of AIRS at those altitudes (Waters et al., 2006). There exists a dry bias of
20% between INSAT-3D and SAPHIR in first two layers but reduced to less than 10% above
(Figure 7e). In this context it is worth to quote findings of Venkat Ratnam et al. (2013) who have
reported that the first layer (1000-850 hPa) of SAPHIR has large difference when compared to
similar satellites. Thus, the present result of large difference between INSAT-3D and SAPHIR in
the lower most layers is expected. Note that no data is there in SAPHIR above 27° due to its low
inclination.
**4. Consistency check in T measurements of INSAT-3D in the UTLS region**
From the previous section, it is clear that INSAT-3D overestimates T by 1% in the UTLS
region. However, in order to check whether this positive bias is consistent or not, we compared
the tropopause temperature obtained from radiosonde. The cold point tropopause temperature
(CPT) which is the minimum in the temperature profile below 20 km is obtained from
radiosonde and INSAT-3D for each day during 2014 and 2015 and is shown in Figure 8.
Consistent positive bias of 2-3 K is seen in CPT between INSAT-3D and radiosonde as expected,
however, general trends match well between the two. The CPT obtained from INSAT-3D
matches well with the radiosonde observations and shows a clear annual variability with higher
values during the summer monsoon months (JJA) and lower values observed in winter months
(DJF). This seasonal variability of the CPT over the Indian Monsoon region during different





seasons is consistent with that reported by Mehta et al. (2010). These results are also consistent

with that reported earlier over other tropical latitudes (Newell et al., 1969; Reed and Vlcek,

1969; Reid and Gage, 1996; Seidel et al., 2000) who attributed it to the annual modulation of

Hadley cell. Thus, INSAT-3D data can be effectively utilized to investigate the tropopause

characteristics, however, with a known caution of overestimation of T by 2-3 K. As the data

from INSAT-3D is available for almost every hour this data is very much useful to investigate

Stratosphere Troposphere Exchange (STE) process occurring at sub-daily scales.

**5. Summary and Conclusions**

The quality of the new data product mainly the temperature and relative humidity obtained

from the sounder payload onboard INSAT-3D is discussed. A detailed comparison of the data

(temperature and relative humidity) obtained from INSAT-3D with the existing in-situ

radiosonde measurements over the entire Indian region, other similar satellite (AIRS, MLS and

SAPHIR) observations and re-analysis (ERA-Interim and NCEP) datasets has been carried out in

the present study. Following are the main conclusions drawn from the study.

1. INSAT-3D provides measurements with very good spatial and temporal coverage over

   the Indian region when compared to any other satellites as expected.

2. INSAT-3D is able to measure the general features of temperature and relative humidity

   similar to the radiosonde observations even at sub-daily scales. However, magnitudes

   differs (underestimates) in relative humidity measured by INSAT-3D. There is no day-

   night difference in the temperature measurements of INSAT-3D.

3. The mean difference between INSAT-3D and radiosonde temperature in the troposphere

   is less than 0.5K with standard deviations of 1K. However, mean difference in RH is as

   high as 20-30% with standard deviations of 40-60%.



4. The RH obtained from INSAT-3D shows high correlation values (0.6-0.7) with the Gadanki radiosonde RH than the IMD radiosonde (less than 0.5) due to improved sensor.

5. There exists consistent positive bias (~ 2-3 K) in temperature in the upper troposphere and lower stratosphere in INSAT-3D.

6. A dry bias of 10-25% in the INSAT-3D measured RH when compared to similar satellites and reanalysis data sets are noticed.

**7.** In general, temperature from INSAT-3D agrees well with all the other satellite measurements and reanalysis data sets below 25$^o$N, whereas a difference of ~4K in temperature above 25$^o$N is noticed. INSAT-3D shows less temperature difference around tropopause region with AIRS and ERA-Interim datasets.

It is found that there exists large difference between INSAT-3D and other datasets both in temperature and relative humidity above 25°N latitude. Thus, caution is advised in using INSAT-3D data over those locations. It is important to note that INSAT-3D shows good agreement with the conventional in-situ radiosonde observations of both Gadanki and IMD locations over the Indian region giving a sign of good reliability to use the former datasets for measuring the temperature and relative humidity spatially and temporally. Very low difference in temperature between INSAT-3D and radiosonde observations provides the scope of using the INSAT-3D data into the numerical weather models for better forecasts. However, caution is again advised while using the RH where most of the time a mean dry bias of 20-30% is noticed. Though consistent positive bias of ~2-3 K is observed in the cold point tropopause temperatures, the variability in tropopause obtained from INSAT-3D shows excellent match with the in-situ radiosonde observations during 2014 and 2015. Thus, INSAT-3D data can also be used to study



the tropopause characteristics at sub-daily scales which are not possible with any existing
satellites and hence Stratosphere-Troposphere Exchange processes.

**Acknowledgements:** The INSAT-3D data used in the present study obtianed from MOSDAC is
greatly acknowldged. We thank AIRS, MLS, ERA-Interim and NCEP teams for providing data
used in the present study through their ftp sites.





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

488





**Figure captions:**

**Figure 1.** Spatial variation of (a) temperature and (b) relative humidity over Indian region at 850 hPa pressure level obtained from INSAT-3D satellite on 2 May 2015 (averaged over a day). The filled circles (magenta) in both the panels show the locations of IMD radiosonde stations selected within 0-40˚N latitude and 60-100˚E longitude (Indian region) for comparing INSAT-3D observations. White patches show the non-availability of the data.

**Figure 2.** Spatial variation of temperature over Indian region at 500 hPa pressure level obtained from (a) INSAT-3D, (b) MLS, (c) AIRS, (d) ERA-Interim and (e) NCEP on 2 May 2015 around 1330 IST. (f) – (j) same as (a) to (e) but for relative humidity. White patches show the non-availability of the data.

**Figure 3:** Temporal variation of (a) temperature and (b) relative humidity obtained from radiosonde launched over Gadanki during the TTD Campaign conducted from 27 Jan. 2014 to 30 Jan. 2014. White patches show the non-availability of the data. (c) and (d) same as (a) and (b) but observed by INSAT-3D. The mean profiles of (e) temperature and (f) relative humidity obtained from radiosonde (red) and INSAT-3D (blue). Horizontal lines indicate standard deviations.

**Figure 4:** Correlation coefficients obtained in (a) temperature and (b) relative humidity at different pressure levels between INSAT-3D and 12 UTC Gadanki radiosondes (red line) and 00 UTC IMD radiosondes (black line). Horizontal bars show the deviations in correlation coefficients obtained from 34 stations. Note that correlation coefficient up to 300 hPa is only obtained for relative humidity.

**Figure 5:** Mean difference (thick line) and standard deviation (dotted lines) observed in the temperature between INSAT-3D and radiosonde launched at around 12UTC over Gadanki



during 2014 and 2015. The blue line in (a) represents the number of radiosondes reaching at
different altitudes with top-right axis.
**Figure 6:** Zonal mean latitudinal difference between the INSAT-3D temperature and (a) AIRS,
(b) MLS, (c) ERA-Interim and (d) NCEP temperatures observed during 2014 and 2015. The
contours whose differences are within 1K are omitted.
**Figure 7:** Zonal mean latitudinal difference between the INSAT-3D RH and (a) AIRS RH, (b)
MLS RH, (c) ERA-Interim RH, (d) NCEP RH and (e) SAPHIR RH observed during 2014 and
2015. White patches show the non-availability of the data. The dotted (thick) line contours
show the negative (positive) differences between INSAT-3D and respective data sets.
**Figure 8:** Time series of cold point tropopause temperatures (CPT) observed over Gadanki
during 2014 and 2015 by INSAT-3D (blue line) and radiosonde (red line) at 12 UTC. These
are the 5-point running average of CPT.


**Table caption:**
**Table 1:** The principal absorbing gases of the Infrared radiation in the atmosphere at different
channels in INSAT-3D with their central wavelengths and their purpose of retrieval.



**Table 1:** The principal absorbing gases of the Infrared radiation in the atmosphere at different
channels in INSAT-3D with their central wavelengths and their purpose of retrieval.

| Detector | Ch. No. | Wavelength(µm) | Principal absorbing gas | Purpose |
|---|---|---|---|---|
| **Long wave** | 1 | 14.67 | $CO_2$ | Stratosphere temperature |
| | 2 | 14.31 | $CO_2$ | Tropopause temperature |
| | 3 | 14.03 | $CO_2$ | Upper-level temperature |
| | 4 | 13.64 | $CO_2$ | Mid-level temperature |
| | 5 | 13.33 | $CO_2$ | Low level temperature |
| | 6 | 12.59 | Water vapor | Total precipitable water |
| **Mid wave** | 7 | 11.98 | Water vapor | Surface temperature, moisture |
| | 8 | 10.99 | Window | Surface temperature |
| | 9 | 9.69 | Ozone | Total ozone |
| | 10 | 7.43 | Water vapor | Low level moisture |
| | 11 | 7.04 | Water vapor | Mid-level moisture |
| | 12 | 6.52 | Water vapor | Upper level moisture |
| **Short wave** | 13 | 4.61 | $N_2O$ | Low level temperature |
| | 14 | 4.54 | $N_2O$ | Mid-level temperature |
| | 15 | 4.48 | $CO_2$ | Upper level temperature |
| | 16 | 4.15 | $CO_2$ | Boundary level temperature |
| | 17 | 4.01 | Window | Surface temperature |
| | 18 | 3.79 | Window | Surface temperature, moisture |






**Figures**

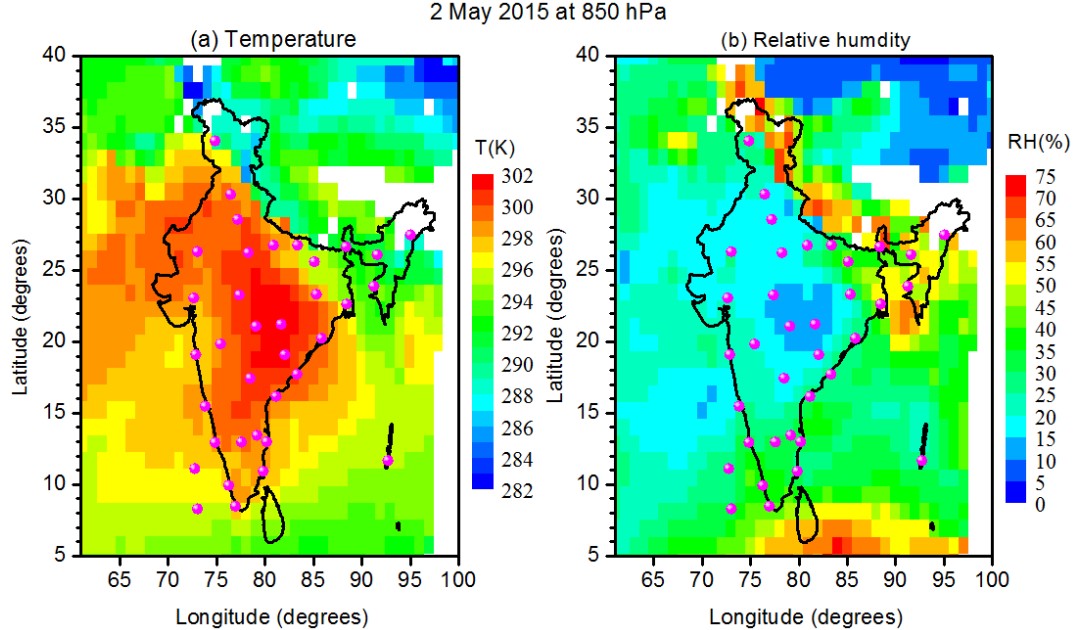

**Figure 1.** Spatial variation of (a) temperature and (b) relative humidity over Indian region at 850

hPa pressure level obtained from INSAT-3D satellite on 2 May 2015 (averaged over a day).

The filled circles (magenta) in both the panels show the locations of IMD radiosonde stations

selected within 0-40 ˚N latitude and 60-100 ˚E longitude (Indian region) for comparing INSAT-

3D observations. White patches show the non-availability of the data.





546

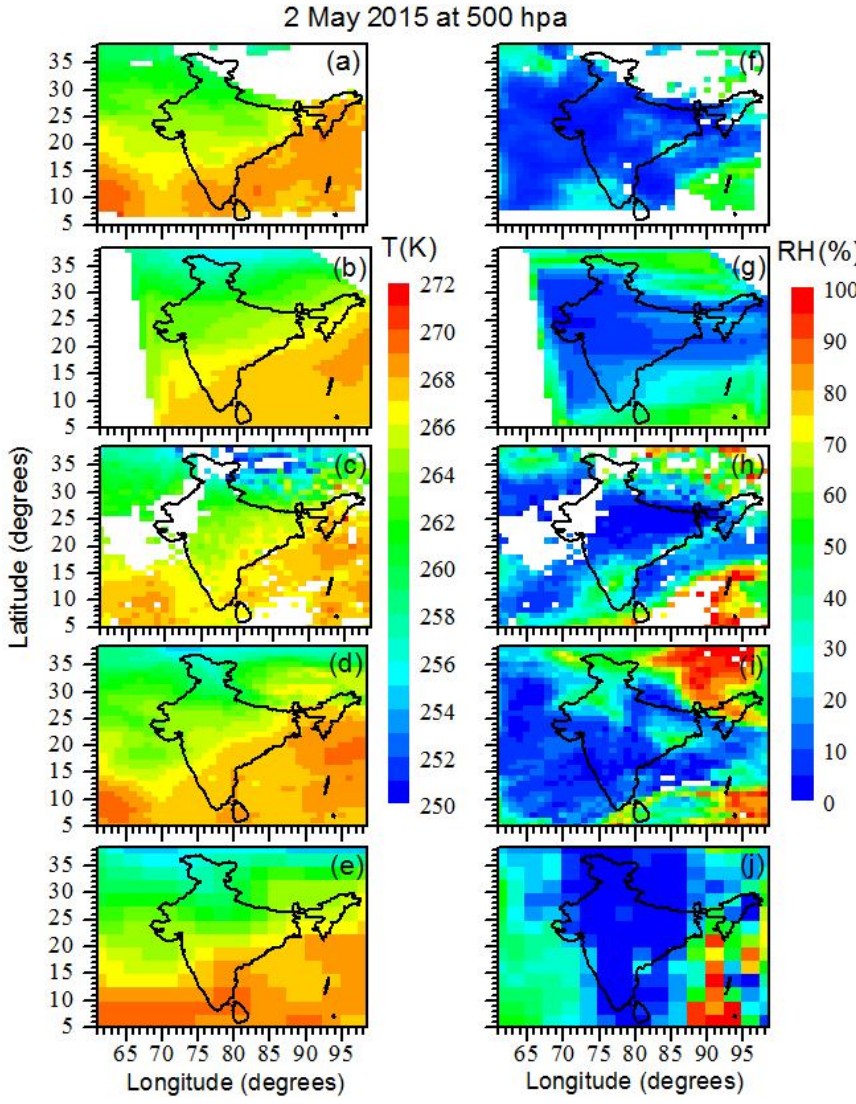

547

**Figure 2.** Spatial variation of temperature over Indian region at 500 hPa pressure level obtained

from (a) INSAT-3D, (b) MLS, (c) AIRS, (d) ERA-Interim and (e) NCEP on 2 May 2015

around 1330 IST. (f) – (j) same as (a) to (e) but for relative humidity. White patches show the

non-availability of the data.

552





553

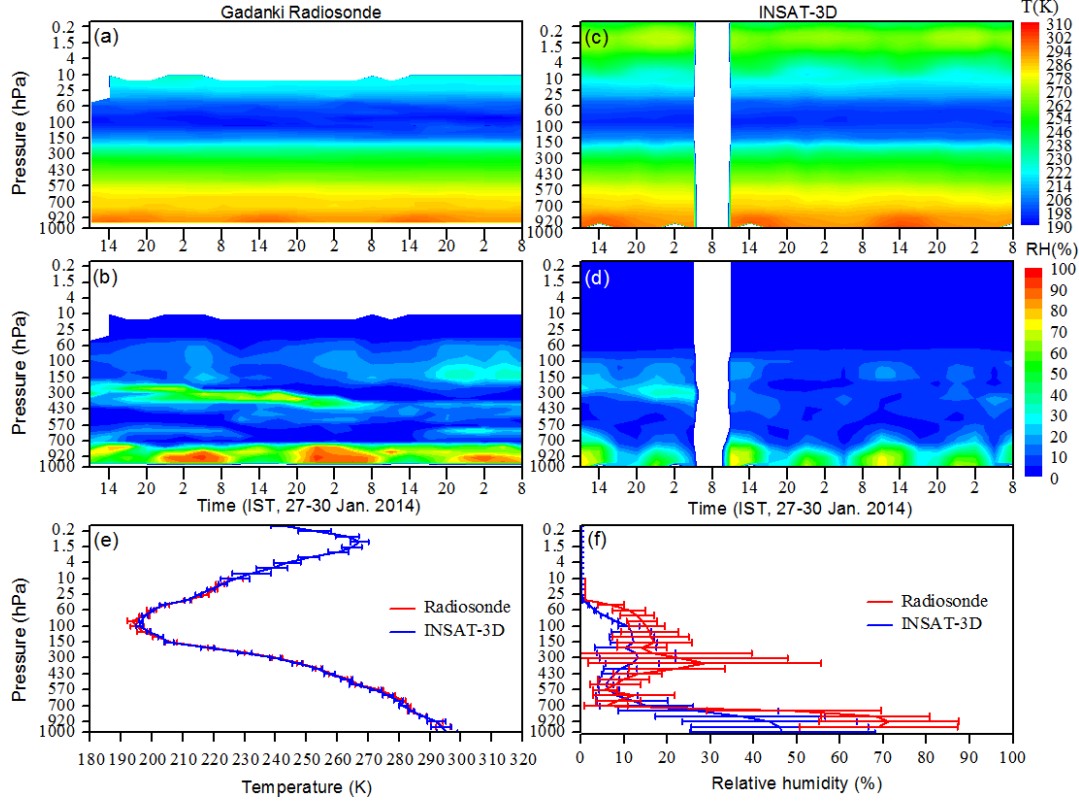

554

**Figure 3:** Temporal variation of (a) temperature and (b) relative humidity obtained from

radiosonde launched over Gadanki during the TTD Campaign conducted from 27 Jan. 2014 to

30 Jan. 2014. White patches show the non-availability of the data. (c) and (d) same as (a) and

(b) but observed by INSAT-3D. The mean profiles of (e) temperature and (f) relative humidity

obtained from radiosonde (red) and INSAT-3D (blue). Horizontal lines indicate standard

deviations.

561





562

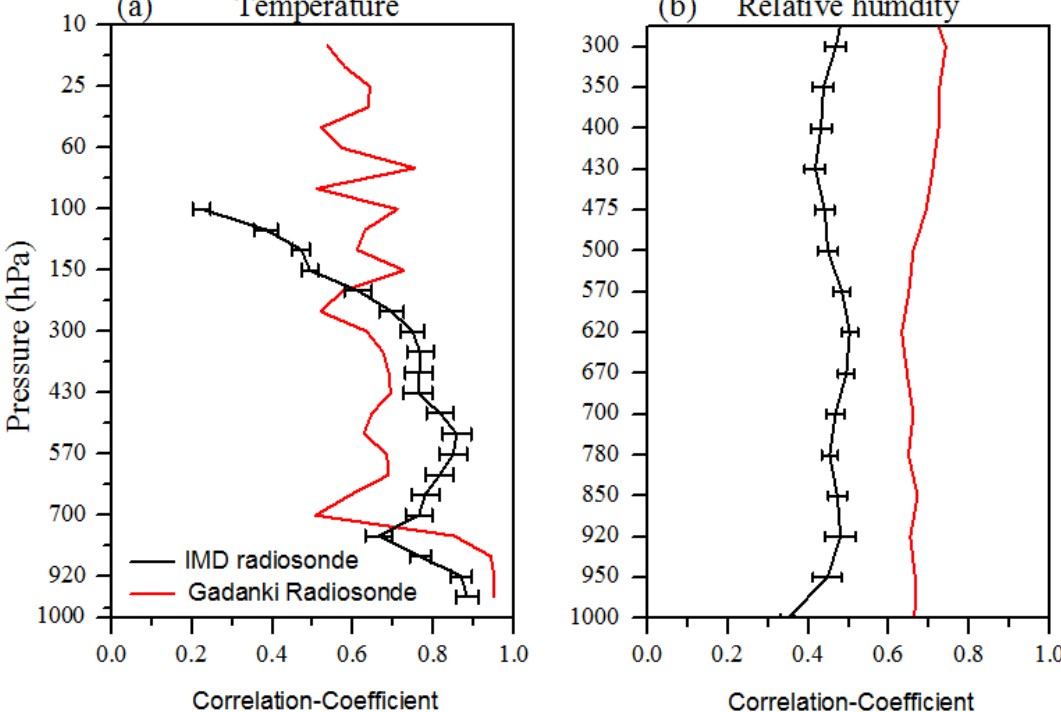

563

**Figure 4:** Correlation coefficients obtained in (a) temperature and (b) relative humidity at different pressure levels between INSAT-3D and 12 UTC Gadanki radiosondes (red line) and 00 UTC IMD radiosondes (black line). Horizontal bars show the deviations in correlation coefficients obtained from 34 stations. Note that correlation coefficient up to 300 hPa is only obtained for relative humidity.

569



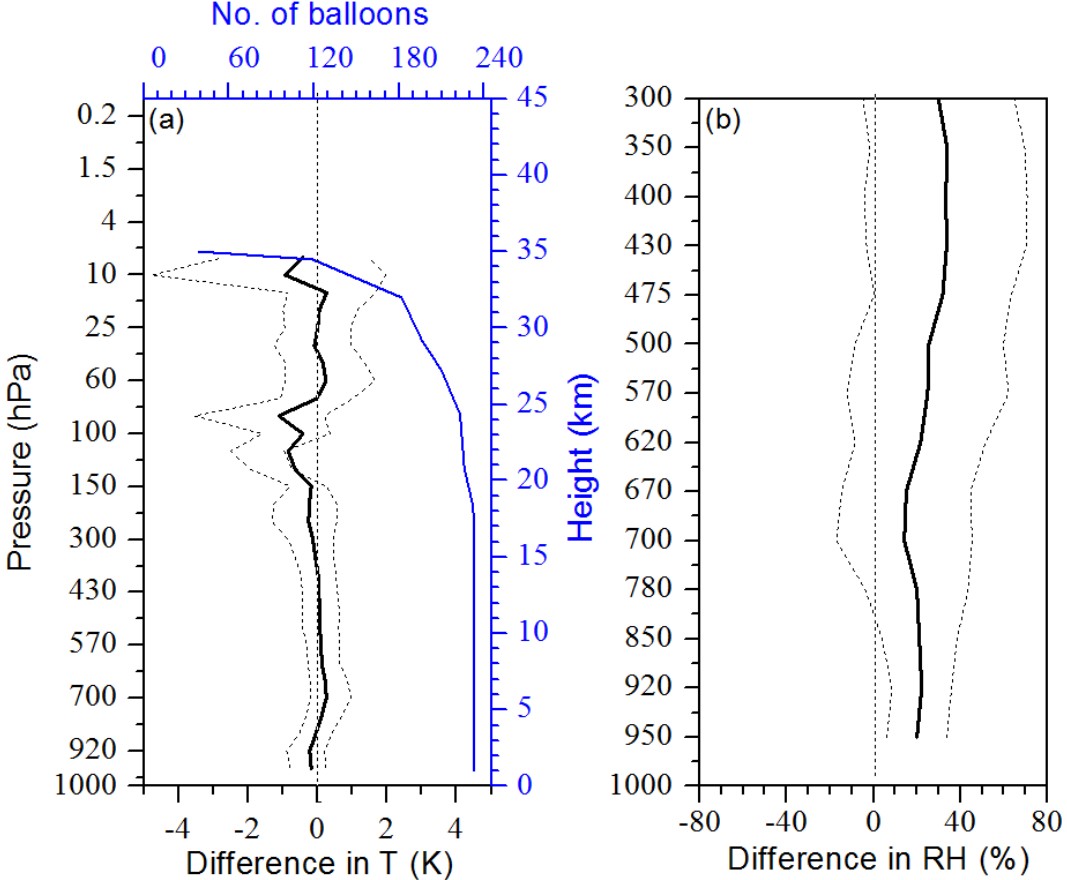

570

**Figure 5:** Mean difference (thick line) and standard deviation (dotted lines) observed in the

temperature between INSAT-3D and radiosonde launched at around 12UTC over Gadanki

during 2014 and 2015. The blue line in (a) represents the number of radiosondes reaching at

different altitudes with top-right axis.







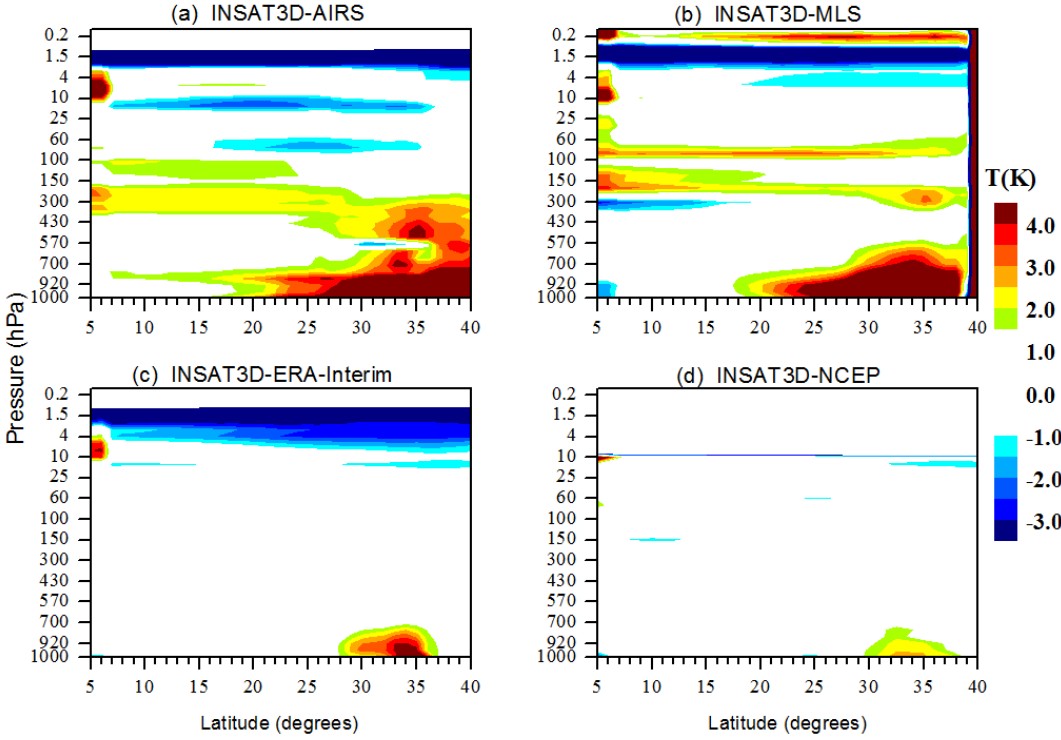


**Figure 6:** Zonal mean latitudinal difference between the INSAT-3D temperature and (a) AIRS,

(b) MLS, (c) ERA-Interim and (d) NCEP temperatures observed during 2014 and 2015. The

contours whose differences are within 1K are omitted.









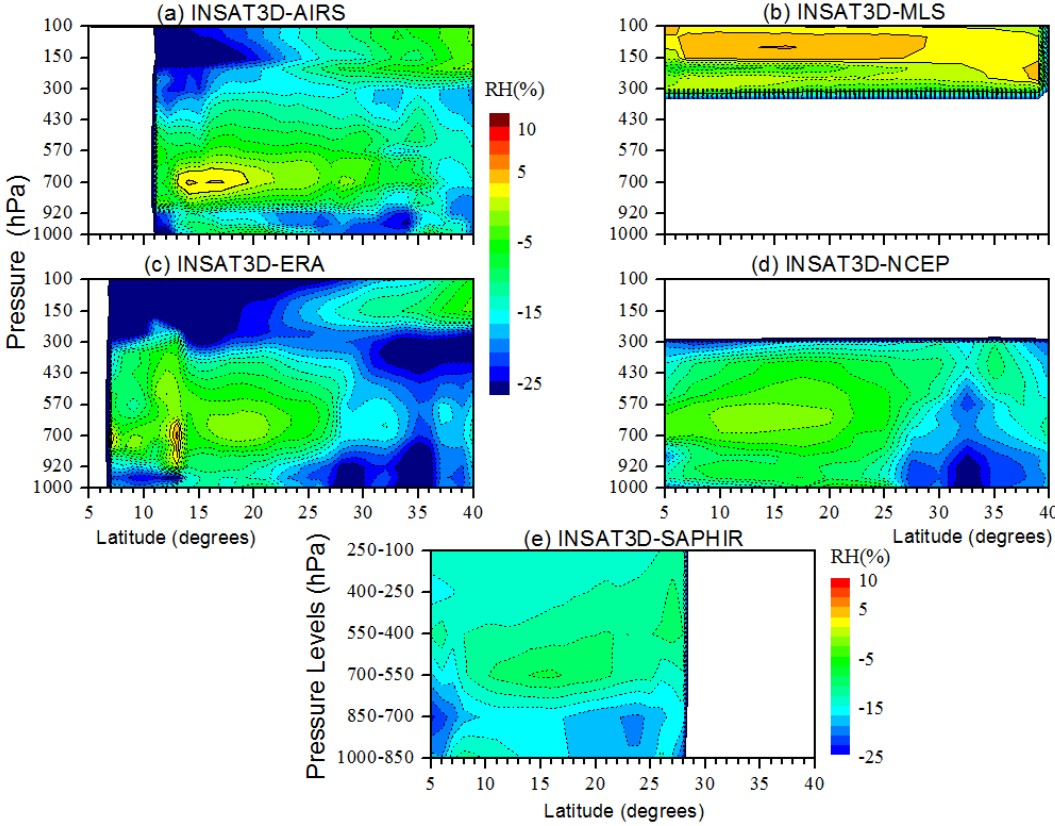


**Figure 7:** Zonal mean latitudinal difference between the INSAT-3D RH and (a) AIRS RH, (b) MLS RH, (c) ERA-Interim RH, (d) NCEP RH and (e) SAPHIR RH observed during 2014 and 2015. White patches show the non-availability of the data. The dotted (thick) line contours show the negative (positive) differences between INSAT-3D and respective data sets.






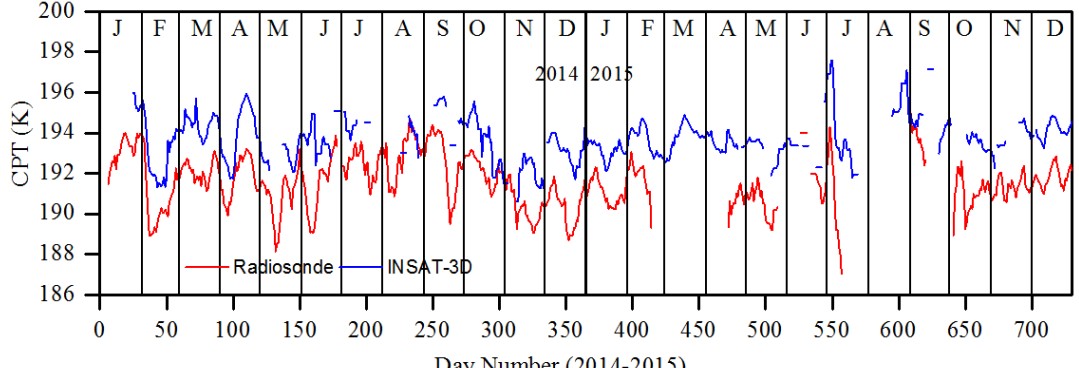


**Figure 8:** Time series of cold point tropopause temperatures (CPT) observed over Gadanki
during 2014 and 2015 by INSAT-3D (blue line) and radiosonde (red line) at 12 UTC. These
are the 5-point running average of CPT.
