# Peer review of "Validation of INSAT-3D sounder data with in-situ measurements and other similar satellite observations over Indian region"

_Atmospheric Measurement Techniques, 2016_

## Referee Comment (RC1) · Anonymous Referee #1 · 18 Aug 2016

**Peer Review**
*Atmos. Meas. Tech. Discuss.*
**Manuscript amt-2016-195**

"Validation of INSAT-3D sounder data with in-situ measurements and other similar satellite observations over Indian region"
by Ratnam, Kumar and Jayaraman

**Overview**

This paper is a straightforward paper on validation of atmospheric profile retrievals from the INSAT-3D geostationary satellite infrared sounder. Because accurate atmospheric profile retrievals from a geostationary sounder are important environmental data records (EDRs) of use to weather forecasting, and because validation of EDRs supports broader calibration/validation of the sensor radiances, this paper is appropriate for eventual publication in *AMT*. However, below are suggestions/comments and questions that should be addressed before publication.

**General Comments**

1. There are problems with English usage/style throughout the manuscript. These are not limited to grammatical errors alone, but include sentence construction, incorrect definite/indefinite article usage, inconsistent tense usage, etc. I do not have time to go through an correct all these problems, so I leave it to the Authors and/or Copy Editor to do this.

2. The Authors choose to present their water vapor validation results in terms of RH, but the do not indicate why that is. RH involves both water vapor and temperature, and thus errors may result from errors temperature in addition to moisture. The Authors may want to consider presenting their results in terms of mixing ratio, specific humidity, layer abundance, etc., or otherwise provide rationale as to why they use RH.

3. A bit more detail is required from the Authors on relevant technical methodologies that were employed. Examples include:
- Mention that both temperature and water vapor profiles are used for calculation of the RH.

- Specifics concerning the space-time radiosonde collocation criteria and/or methodology are needed. What are the collocation distances in terms of space and time?

**Specific Comments, Suggestions and Questions**

- Lines 107–109: In discussion of the retrieval algorithm, the Authors needs to also make specific mention of the cloud mask / cloud clearing algorithm that is being used. Also, the Authors should provide some indication of the corresponding yield of their product (after cloud masking/clearing).

- Line 114: $\pm 2\sigma$ of what?

- Line 119: "high altitude resolution" — what does this terminology mean?

- Lines 128–129, 240–241, 261–262: Interpolating high resolution radiosonde measurements to the sparse pressure levels of the INSAT-3D is not a good way of doing this — among other things, the radiosonde has fine scale structure information in it that may be erroneously aliased or missed in a simple interpolation to a sparse sampling interval. The Authors should see Nalli et al. (2013) for more information on this (see bibliographic information below). While it is not required that the Authors employ the approach in the reference cited, they should nevertheless employ a more rigorous approach than simple interpolation.

- Lines 206–207: The Authors should specifically state how they are calculating RH in this paper. They don't need to provide the equation, but they need to make it clear that they are using both temperature and water vapor profiles.

[Figure]

- Lines 230–231: The Authors should cite some of the prior validation work that was done for other sounder systems (e.g., AIRS, referenced in the paper, and GOES) . To do this, they can simply extend this sentence to read: "It is well known that the most common and widespread in-situ instruments for providing accurate profiles of T and RH are radiosondes, and these are typically employed for satellite sounder validation (e.g., Fetzer et al., 2003; Xie et al., 2013; Nalli et al., 2013)." Bibliographic information for these papers is provided below.

- Lines 251–254: Do you know where these moisture biases are originating from? Is this an artifact of using RH instead of specific humidity, etc.? (Per General Comment #2 above).

- Lines 266–267: Authors should provide reason why they only go up to 300 hPa for RH.

- Line 283: What is meant by "fractional difference" in T? The figure shows the mean difference.

- Lines 290–291: "However, positive bias. . . (shown as standard deviations). . ." doesn't make sense. Standard deviations do not measure systematic error.

- Lines 294–296: "Most striking feature to be noticed is the consistent positive bias of 1% ( 2 K) in T. . ." — I do not see this in the figure.

- Lines 297–298: "Standard deviations show dry bias. . ." — again, this doesn't make sense to me — how do standard deviations show systematic error?

- Lines 298–300: Do the Authors have any ideas of what causes the "huge dry bias in RH"?

- Lines 319–321: The Authors should provide some explanations as to what's may be leading to these findings.

- Line 331: "integrated relative humidity" — what is that?

- Line 390: "below $25°$N" is this due to observing geometry? The Authors should provide a reason.

- Line 411, References: The Authors should include the references provided at the bottom of the review.

**Technical Corrections**

- Line 25 (and elsewhere in the paper): "12 UTC" — please include the corresponding local time when it is germane to the discussion (i.e., in this case the authors emphasize that these are evening soundings, thus we need to know the local time to appreciate what specific time in the evening).

- Lines 138–139: The local equator crossing times for Aqua are 01:30 and 13:30 local time (LT), *not* UTC.

- Line 200: I recommend simply deleting this section header — it isn't necessary.

- Line 225: With the above deletion of the previous section header, this section header now becomes 3.1, and the section header on Line 256, originally mislabeled as 3.2, can remain 3.2.

- Lines 244–245: The sentence "White patches…" may be deleted as it is already stated in the Figure caption.

- Line 301: Section header should be 3.3 (not 3.2).

- Line 302: Replace "measured" with "retrieved".
- Lines 313–314: Delete sentence "Note that the differences that are..." since this is already mentioned in the figure caption.

- Line 383: Replace "shows high correlation values" with "were well correlated".

- Lines 512–513: The figure caption is missing the text describing subplot (b).

**References**

Fetzer, E., et al., 2003: AIRS/AMSU/HSB validation. *IEEE Trans. Geosci. Remote Sensing*, **41 (2)**, 418–431.

Nalli, N. R., et al., 2013: Validation of satellite sounder environmental data records: Application to the Cross-track Infrared Microwave Sounder Suite. *J. Geophys. Res. Atmos.*, **118**, 1–16, doi:10.1002/2013JD020436.

Xie, H., et al., 2013: Integration and ocean-based prelaunch validation of GOES-R Advanced Baseline Imager legacy atmospheric products. *J. Atmos. Ocean. Tech.*, **30 (8)**, 1743–1756, doi:{10.1175/JTECH-D-12-00120.1}.

——————————————————

---

## Referee Comment (RC2) · J.M. Blaisdell (Referee) · 2 Sep 2016

This paper provides a good summary of validation testing done of INSAT-3D sounder products. The authors clearly describe comparisons they have made and indicate validity of the product for weather applications. They also forthrightly discuss limitations of the product in some spatial and vertical regions. There is one obvious error in either the comparison product used from AIRS or in the description of it. There are also some errors in the formatting of the references, and a few other technical corrections. I recommend publication of this paper after these minor issues are addressed.

The abstract and introduction are well-written and comparisons with satellite data sets and radiosondes are well described. We always want more validation, but as report of

validation to date this paper is entirely adequate. The conclusions seem supported by the data presented.

The one obvious error is in Figure 2(c) and 2(h). This image is described as the AIRS Level 3 daily gridded product for May 2, 2015. This cannot be correct as the AIRS daily gridded product will have orbit gaps at this latitude. (On this date, the orbit gap crosses Sri Lanka and central India.) Perhaps this is an 8-day or monthly gridded product instead, or has somehow been filled? If it is not the daily product, that would explain why it differs most from the other products shown. The authors should either replace this image with the correct daily gridded product or explain what product they actually used. This will not affect the substance of the paper.

The authors could strengthen their argument for INSAT-3D in the introduction by pointing out incomplete coverage of AIRS and MLS because of orbit gaps in tropical regions where geostationary sounder has complete coverage.

Two questions come to mind for further research, which the authors do NOT need to address in this paper:

1) If temperature bias correction is made as suggested, how much improvement is made in relative humidity?

2) AIRS and MLS data could be better used by using actual time of level 2 observation to compare with INSAT sounding closest in time, since INSAT is available every hour and AIRS and MLS vary by up to 2 hours in local observation time because of looking to side.

Technical corrections to figures:

Figure 5 label (a) and (b)

Figure 6 is labeled (a b c c) should be (a b c d)

Technical corrections to references:

Aumann has names incorrect

Mitra has names in wrong format

TIAN is incomplete and in the middle of Susskind

Spelling corrections in text:

line 18, tropics not topics; line 282, quantify not quantity; line 294, overcast not over; line 299, provide not provided
* * *

---

## Author Comment (AC1) · 26 Oct 2016

Replies to Reviewer#1 comments/suggestions

Overview This paper is a straightforward paper on validation of atmospheric profile retrievals from the INSAT-3D geostationary satellite infrared sounder. Because accurate atmospheric profile retrievals from a geostationary sounder are important environmental data records (EDRs) of use to weather forecasting, and because validation of EDRs supports broader calibration/validation of the sensor radiances, this paper is appropriate for eventual publication in AMT. However, below are suggestions/comments and questions that should be addressed before publication.

[Figure]

Reply: First of all we wish to thank the reviewer for going through the manuscript carefully and offering potential solutions for the betterment.

General Comments There are problems with English usage/style throughout the manuscript. These are not limited to grammatical errors alone, but include sentence construction, incorrect definite/indefinite article usage, inconsistent tense usage, etc. I do not have time to go through and correct all these problems, so I leave it to the Authors and/or Copy Editor to do this.

Reply: We have taken utmost care in minimizing grammatical mistakes and wrong English usage in the revised manuscript.

The Authors choose to present their water vapor validation results in terms of RH, but they do not indicate why that is. RH involves both water vapor and temperature and thus errors may result from errors temperature in addition to moisture. The Authors may want to consider presenting their results in terms of mixing ratio, specific humidity, layer abundance, etc., or otherwise provide rationale as to why they use RH.

Reply: Thanks for raising this issue. In the revised manuscript we have used mixing ratio instead of RH for INSAT-3D and all other satellite measurements except SAPHIR. Kindly note that we have RH only from this satellite.

A bit more detail is required from the Authors on relevant technical methodologies that were employed. Examples include: Mention that both temperature and water vapor profiles are used for calculation of the RH. Specifics concerning the space-time radiosonde collocation criteria and/or methodology are needed. What are the collocation distances in terms of space and time?

Reply: We have included details of above mentioned issues in the revised manuscript. Note that we have used mixing ratio in the revised manuscript. We also included details of the collocation criteria.

Specific Comments, Suggestions and Questions Lines 107–109: In discussion of the

retrieval algorithm, the Authors needs to also make specific mention of the cloud mask / cloud clearing algorithm that is being used. Also, the Authors should provide some indication of the corresponding yield of their product (after cloud masking/clearing).

Reply: Suitable reference for the cloud masking algorithm used for INSAT-3D along with brief details are provided in the revised manuscript.

Line 114: $\pm 2\sigma$ of what?

Reply: $\pm 2\sigma$ of the parameter (here temperature and water vapor).

Line 119: "high altitude resolution" — what does this terminology mean?

Reply: It means high vertical space resolution. Clarified in the revised manuscript.

Lines 128–129, 240–241, 261–262: Interpolating high resolution radiosonde measurements to the sparse pressure levels of the INSAT-3D is not a good way of doing this — among other things, the radiosonde has fine scale structure information in it that may be erroneously aliased or missed in a simple interpolation to a sparse sampling interval. The Authors should see Nalli et al. (2013) for more Information on this (see bibliographic information below). While it is not required that the Authors employ the approach in the reference cited, they should nevertheless employ a more rigorous approach than simple interpolation.

Reply: Thanks for nice suggestion. However, note that except Gadanki radiosonde measurements, no other measurements used in the present study are of high vertical resolution. Thus, we have retained same analysis procedure as was done in our earlier study (Venkat Ratnam et al., 2013) while referring to the Nalli et al., (2013) in the revised manuscript.

Lines 206–207: The Authors should specifically state how they are calculating RH in this paper. They don't need to provide the equation, but they need to make it clear that they are using both temperature and water vapor profiles.

[Figure]

Reply: As mentioned earlier, note that we have used mixing ratio instead of RH in the revised manuscript. As suggested, INSAT-3D water vapor mixing ratio is now compared with the mixing ratios obtained from AIRS, MLS, ERA-Interim and NCEP re-analysis data instead of RH. But, the mixing ratio of INSAT-3D is converted to RH so as to compare with SAPHIR and explicitly mentioned that both temperature and water vapor profiles are used for RH estimation as suggested.

Lines 230–231: The Authors should cite some of the prior validation work that was done for other sounder systems (e.g., AIRS, referenced in the paper, and GOES). To do this, they can simply extend this sentence to read: "It is well known that the most common and widespread in-situ instruments for providing accurate profiles of T and RH are radiosondes, and these are typically employed for satellite sounder validation (e.g., Fetzer et al., 2003; Xie et al., 2013; Nalli et al., 2013)." Bibliographic information for these papers is provided below.

Reply: We thank the reviewer for providing updated references which are quoted at suitable places in the revised manuscript.

Lines 251–254: Do you know where these moisture biases are originating from? Is this an artifact of using RH instead of specific humidity, etc.? (Per General Comment #2 above).

Reply: In the revised manuscript we have used mixing ratio and could see similar result. Thus, the differences observed in the present study are not an artifact. At this moment we do not know the exact reason.

Lines 266–267: Authors should provide reason why they only go up to 300 hPa for RH.

Reply: In general, the RH obtained from radiosonde are accurate below -40oC that is $\sim$12 km and beyond this altitude the radiosonde humidity sensors are not sensitive.

Line 283: What is meant by "fractional difference" in T? The figure shows the mean difference.

Reply: Fractional difference in temperature and water vapor measurements is the difference in the temperature or water vapor values between INSAT-3D and the comparing instrument to the values (T or RH) of INSAT-3D.The formula is given below.

where $\Delta$ is the fractional difference of the atmospheric parameter (A (Temperature, Relative humidity)) between the Radiosonde and INSAT 3D and its value ranges from -1 to 1.

Lines 290–291: "However, positive bias. . . (Shown as standard deviations). . . " doesn't make sense. Standard deviations do not measure systematic error. Reply: Corrected. We mean the mean difference not the standard deviations.

Lines 294–296: "Most striking feature to be noticed is the consistent positive bias of 1% ( 2 K) in T. . . " — I do not see this in the figure

Reply: The fractional difference plotted as calculated from the equation below

The percentage of the fractional difference of temperature in the upper troposphere is 1% (which means the fractional difference value is 0.01). The difference in the temperature between INSAT and Radiosonde can be estimated.

Say R is the temperature value obtained from radiosonde and if x is the fractional difference value then we can obtain the temperature of INSAT-3D by using the formula I= R(1-x); Eg R=190K; x(fractional difference)=0.01; I=190(1-0.01);I=190-190*0.01; I=190-1.9; I=188.1K ; The difference between I and R is $\sim$2K

Lines 297–298: "Standard deviations show dry bias. . . " — again, this doesn't make sense to me — how do standard deviations show systematic error?

Reply: Corrected. We meant the mean difference not the standard deviations.

Lines 298–300: Do the Authors have any ideas of what causes the "huge dry bias in RH"?

Reply: We really do not the reasons at this point.

Lines 319–321: The Authors should provide some explanations as to what's may be leading to these findings.

Reply: In general, INSAT-3D data has been assimilated in model forecast and we advise caution in using this data. In this way, this study is helpful providing caution before making forecasting using this data.

Line 331: "integrated relative humidity" — what is that?

Reply: The SAPHIR provides layer-averaged RH values in six pressure layers (1000–850, 850–700, 700–550, 550–400, 400–250 and 250–100 hPa).

Line 390: "below 25" is this due to observing geometry? The Authors should provide a reason.

Reply: We really do not the reasons at this point.

Line 411, References: The Authors should include the references provided at the bottom of the review.

Reply: Thanks for updating us and we have included all the suggested additional references in the revised manuscript.

Technical Corrections

Line 25 (and elsewhere in the paper): "12 UTC" — please include the corresponding local time when it is germane to the discussion (i.e., in this case the authors emphasize that these are evening soundings, thus we need to know the local time to appreciate what specific time in the evening).

Reply: Added.

Lines 138–139: The local equator crossing times for Aqua are 01:30 and 13:30 local time (LT), not UTC.

Reply: Corrected.

Line 200: I recommend simply deleting this section header — it isn't necessary.

Reply: Deleted.

Line 225: With the above deletion of the previous section header, this section header now becomes 3.1, and the section header on Line 256, originally mislabeled as 3.2, can remain 3.2

Reply: Corrected.

Lines 244–245: The sentence "White patches. . . " may be deleted as it is already stated in the Figure caption

Reply: Deleted.

Line 301: Section header should be 3.3 (not 3.2).

Reply: Corrected.

Line 302: Replace "measured" with "retrieved".

Reply: Replaced.

Lines 313–314: Delete sentence "Note that the differences that are. . . " since this is already mentioned in the figure caption.

Reply: Deleted.

Line 383: Replace "shows high correlation values" with "were well correlated".

Reply: Replaced.

Lines 512–513: The figure caption is missing the text describing subplot (b).

Reply: Added.

References Fetzer, E., et al., 2003: AIRS/AMSU/HSB validation. IEEE Trans. Geosci. Remote Sensing, 41 (2), 418–431. Nalli, N. R., et al., 2013: Validation of satellite

sounder environmental data records: Application to the Cross-track Infrared Microwave Sounder Suite. J. Geophys. Res. Atmos., 118, 1–16, doi:10.1002/2013JD020436. Xie, H., et al., 2013: Integration and ocean-based prelaunch validation of GOES-R Advanced Baseline Imager legacy atmospheric products. J. Atmos. Ocean. Tech., 30 (8), 1743–1756,

The above references are added in the revised manuscript.

We once again thank the reviewer for his/her constructive comments/suggestions which made us to improve the manuscript content significantly.

—END—